# Spatially conserved motifs in complement control protein domains determine functionality in regulators of complement activation-family proteins

Hina Ojha[1], Payel Ghosh[2], Hemendra Singh Panwar [1], Rajashri Shende[1], Aishwarya Gondane[2], Shekhar C. Mande [3,4] & Arvind Sahu [1]

Regulation of complement activation in the host cells is mediated primarily by the regulators of complement activation (RCA) family proteins that are formed by tandemly repeating complement control protein (CCP) domains. Functional annotation of these proteins, however, is challenging as contiguous CCP domains are found in proteins with varied functions. Here, by employing an in silico approach, we identify five motifs which are conserved spatially in a specific order in the regulatory CCP domains of known RCA proteins. We report that the presence of these motifs in a specific pattern is sufficient to annotate regulatory domains in RCA proteins. We show that incorporation of the lost motif in the fourth long-homologous repeat (LHR-D) in complement receptor 1 regains its regulatory activity. Additionally, the motif pattern also helped annotate human polydom as a complement regulator. Thus, we propose that the motifs identified here are the determinants of functionality in RCA proteins.

[1] Complement Biology Laboratory, National Centre for Cell Science, S. P. Pune University campus, Pune 411007, India. [2] Bioinformatics Centre, S. P. Pune University, Pune 411007, India. [3] Structural Biology Laboratory, National Centre for Cell Science, S. P. Pune University campus, Pune 411007, India. [4] Present address: Council of Scientific and Industrial Research (CSIR), Anusandhan Bhawan, 2 Rafi Marg, New Delhi 110001, India. Correspondence and requests for materials should be addressed to A.S. (email: arvindsahu@nccs.res.in)

The complement system is a key constituent of innate immunity, which is believed to have appeared in evolution at least 600 million years ago[1]. It functions as a surveillance system in the body—facilitates pathogen elimination directly via lysis, and indirectly via enhancing phagocytosis and contributing to activation of adaptive immunity[2,3]. Triggering of complement occurs via three major pathways namely classical, alternative and lectin pathways, which converge with the formation of C3-cleaving enzymes C3 convertases (C4b2a and C3bBb) on the pathogen surface. Protection of host cells from complement is largely mediated by a family of proteins termed regulators of complement activation (RCA), which target C3 convertases. It is, therefore, not surprising that mutations and polymorphisms in RCA proteins are linked to various diseases such as age-related macular degeneration, atypical haemolytic uraemic syndrome and dense deposit disease[4–6].

In humans, RCA proteins cluster on the chromosome 1q32[7]. The notable members of this family that effectively regulate complement include decay-accelerating factor (DAF; CD55), membrane cofactor protein (MCP; CD46), complement receptor 1 (CR1; CD35), C4b binding protein (C4BP) and factor H (FH)[8]. The two regulatory activities owing to which these proteins regulate C3 convertases are termed decay-accelerating activity (DAA) and cofactor activity (CFA). DAA refers to irreversible dissociation of C3 convertases by the RCA protein and CFA refers to inactivation of the non-catalytic subunit (C3b/C4b) of C3 convertases by the serine protease factor I due to its recruitment onto the C3b/C4b-RCA protein complex.

The complement system is known to provide effective protection against various pathogens including viruses. Notably, the system is capable of deftly recognizing and neutralizing viruses. Thus, to escape the complement attack, viruses employ various subversion mechanisms. Interestingly, the large DNA viruses such as orthopox and herpesviruses, encode mimics of RCA proteins to protect themselves from the host complement[9]. Like human RCA proteins, such mimics also possess DAA and CFA. The important examples of viral RCA proteins are: VCP (vaccinia virus complement regulator), SPICE (smallpox complement regulator), MOPICE (monkeypox virus complement regulator), Kaposica (HHV-8 complement regulator), HVS-CCPH (Herpesvirus saimiri complement regulator) and RCP-1 (Rhesus rhadinovirus complement regulator)[10].

A characteristic feature of the RCA proteins is the presence of concatenated complement control protein (CCP) modules (also known as sushi domains), which are linked by short linkers of 3–8 amino acids (aa). These CCP modules are composed of ~60–70 aa with four invariant cysteines forming disulfide bonds between $C^I$–$C^{III}$ and $C^{II}$–$C^{IV}$. Earlier sequence analysis showed that most, but not all, CCP modules contain a motif "hXhGXXhXhXC$^{II}$XXG↑hXhXG", where ↑ represents a site of insertion in larger CCP modules[11]. The number of CCP modules in human RCA proteins vary from 4 to 59 (DAF and MCP, 4 CCPs; FH, 20 CCPs; C4BP, 59 CCPs; CR1, 30 CCPs). It is, however, important to mention here that the presence of CCP domains is not restricted to RCA proteins alone, but are also found in a variety of other proteins such as those involved in complement activation, cell adhesion, coagulation, neuro-transmission, cytokine signalling and blood clotting[11].

A wealth of mutagenesis data exists for RCA proteins. Deletion mutagenesis showed that a minimum of 3–4 successive CCP domains in RCA proteins contribute to complement regulation and cell protection from complement-mediated damage[8,12,13]. Site-directed mutagenesis, initially on CR1[14,15], DAF[16], MCP[17] and C4BP[18], and later on viral RCAs[9,19–22] revealed that functional sites reside in each of the functional domains. A major advance on the molecular basis of interaction of RCA proteins

with C3b, however, came more recently owing to the availability of structures of CR1, DAF, MCP, FH and SPICE in complex with C3b[23,24]. Further, the structure of FH in complex with C3b and factor I has also been solved[25]. These structures show that all the RCA proteins bind in an extended orientation to C3b and share the same binding platform suggesting they share common attributes. An apparent conundrum, therefore is, what common attributes annotate a string of CCP domains in an RCA protein as complement-regulatory domains? Knowing this is crucial as it can be employed to classify unannotated regulatory RCA proteins and locate regulatory domains within them.

In the present study, by performing in silico analysis of complement-regulatory domains of known RCA proteins, we identify five motifs which are located in these domains in a specific pattern. Additionally, we also experimentally establish that the identified motif pattern can indeed recognize the regulatory CCP domains. We show here that the motif pattern containing human polydom is a complement regulator, and that incorporation of the lost motif in the fourth long-homologous repeat (LHR-D) in CR1 imparts regulatory function to this repeat. Thus, we present an in silico method to annotate regulatory function to uncharacterized RCA sequences.

## Results

**Complement regulatory CCP domains harbour a signature motif pattern.** Identification of motif(s) that discriminate between complement regulatory and non-regulatory CCP domains demanded a large input dataset comprising of regulator-like sequences. Although genome sequencing of various animals and viruses have generated an enormous amount of sequence data of CCP-containing proteins, only a few sequences have been annotated to have complement regulatory activities. Thus, the first step was to create a dataset of regulator-like RCA sequences. For this, we retrieved RCA-like sequences from NCBI and Uni-Prot, and constructed phylogenetic trees using the Neighbour-Joining method. Thereafter, only sequences that showed evolutionary coupling with functionally characterized complement regulatory proteins were selected for the dataset (Supplementary Figs. 1 and 2; detailed in "Methods"). This eliminated CCP domain-containing human CR2-like and vaccinia virus B5R-like sequences that lack complement regulatory activities. Besides, to prevent biased interpretation and redundancies, sequences that showed >95% similarity were eliminated (Fig. 1a and Supplementary Data 1). Next, from these sequences, we extracted the sequences of the regions that are expected to encompass complement regulatory activity. In particular, we extracted the sequences of three consecutive CCP domains. The rationale for this is that in most regulators, the smallest structural unit that displays regulatory function is formed by three CCPs and the fourth CCP, when required, plays only a supportive role[22,26–28].

Next, we examined the dataset sequences to predict the putative nature of motifs associated with complement regulatory CCP domains. The sequences showed a very few insertions and deletions, if any, suggesting that motifs are likely to be un-gapped. Further, it is also known that the regulatory CCPs interact with multiple proteins to impart regulatory activities. For example, they interact with C3b/C4b and factor I to impart CFA, and with C3b/C4b and C2a/Bb to impart DAA. This, therefore, suggested that regulatory sequences are expected to encompass multiple motifs. Consequently, we chose to employ Multiple Em for Motif Elicitation (MEME) for detection of motifs. Optimization of various input parameters (detailed in "Methods") resulted in the identification of five conserved motifs viz., M1, M2, M3, M4 and M5. These motifs showed the occurrences of many residues with high probability. Importantly, examination of the existing

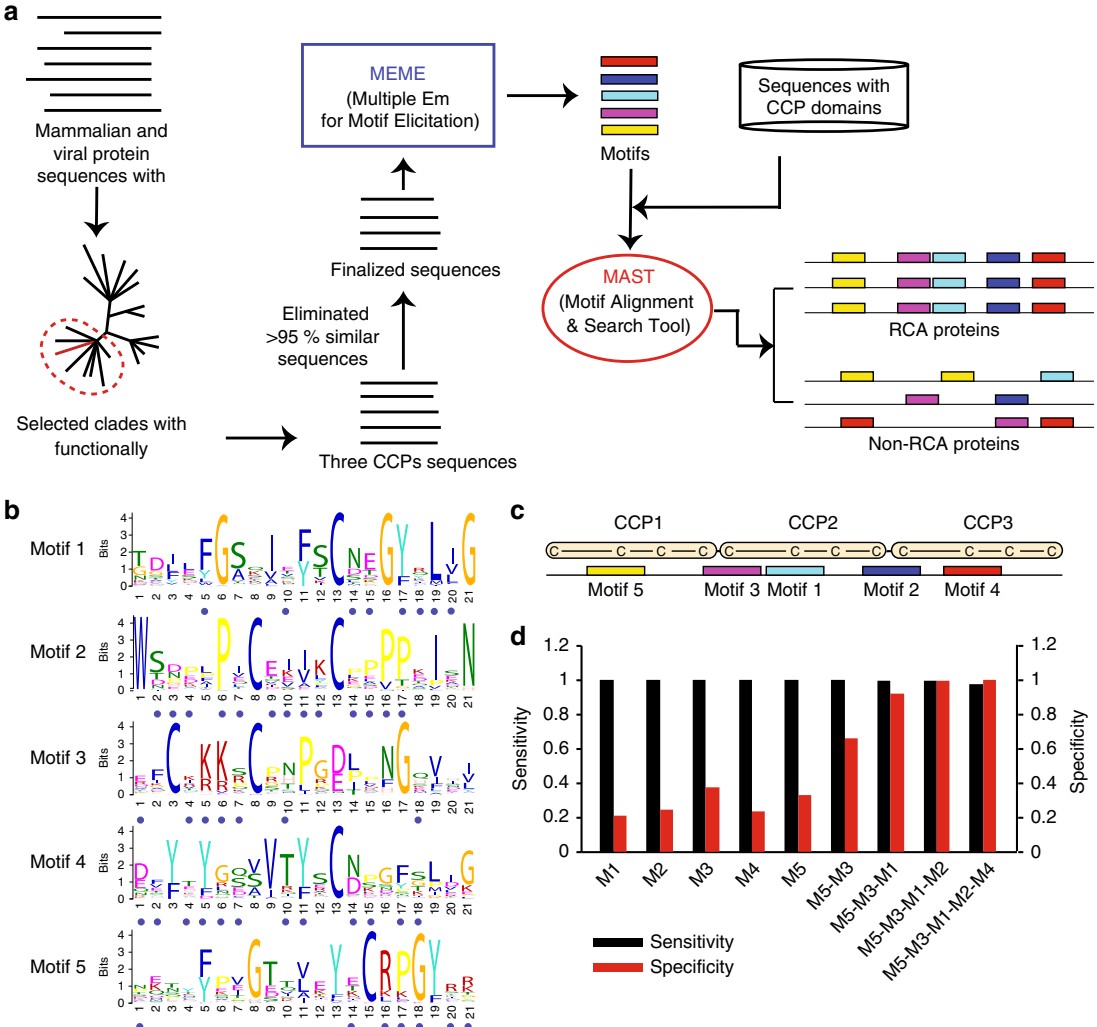

**Fig. 1** A specific pattern of motifs annotate complement regulatory CCP domains. **a** Schematic workflow for motif recognition and annotation of complement regulatory CCPs. The putative complement regulatory mammalian and viral RCA protein sequences were shortlisted by phylogenetic tree construction of CCP domain sequences and selection of clade(s) harbouring at least one functionally characterized RCA protein; <95% similar RCA sequences were considered for identification of motifs with MEME. The MAST-mediated motif scanning using five motifs was employed to identify signature motif patterns in complement regulatory CCPs verses non complement regulatory CCPs. **b** Motifs (M1–M5) generated by MEME. The height of each letter represents the probability of amino acids at the respective position. The blue dot represents the crucial position in regulators according to the mutagenesis data. **c** Order and position of the five motifs as numbered by their *E*-value in a minimal functional unit of three CCPs. The motifs occur in the sequence of Motif 5 (M5—yellow; *E*-value: 2.3e-706), Motif 3 (M3—pink; *E*-value: 2.1e-743), Motif 1 (M1—sky blue; *E*-value: 2.6e-887), Motif 2 (M2—blue; 1.6e-825) and Motif 4 (M4—red; 3.8e-735). **d** Sensitivity and specificity of individual motifs and signature motif patterns

mutagenesis data on RCA proteins revealed that multiple motif residues with high probability are indeed critical for the regulatory activities of these proteins (Fig. 1b; residues marked with a blue dot; Supplementary Data 2).

The MAST scanning showed that the five motifs we identified were present in complement regulatory as well as non-regulatory CCP sequences, but the presence of all the motifs in a specific order (M5-M3-M1-M2-M4) was found only in regulatory sequences (Fig. 2 and Supplementary Fig. 3). Further, the location of the motifs in the regulatory sequences was also completely conserved. The M5, M1 and M4 were positioned around the second Cys of each CCP domain, while the M3 and M2 spanned the linkers (Fig. 1c). Calculation of sensitivity and specificity of motifs showed that though individual motifs display high sensitivity (successful detection in regulatory sequences), high specificity (detection only in regulatory sequences) required the presence of at least 4 motifs (Fig. 1d). Together, these results revealed the presence of a conserved motif pattern in regulatory CCP sequences.

**Signature motif pattern help identify regulatory CCP domains**. To assess whether the identified signature motif pattern is capable of annotating the regulatory domains in RCA proteins, we examined their specific location in human (DAF, MCP, C4BP, CR1 and FH) as well as viral (VCP, SPICE, MOPICE, KAPO-SICA, HVS-CCPH and RCP-1) RCA proteins, some of which harbour multiple non-regulatory CCP domains in addition to the regulatory CCP domains. Intriguingly, in all the examples, MAST scanning with five motifs precisely identified the regulatory domains (Fig. 2). For example, in DAF, CCP2-4 domains impart DAA, and the motif pattern maps only to these domains. Similarly, in CR1, the regulatory activities are imparted by the first 3 CCPs of each of its long-homologous repeats (LHRs; LHR-A, CCP1-7; LHR-B, CCP8-14; LHR-C, CCP15-21; LHR-D, CCP22-28), except LHR-D (i.e., CCP22-24), and the motif pattern maps precisely to these domains (CCP1-3, CCP8-10 and CCP15-17). Notably, the motif pattern was seen in the CCP domains that impart any of the regulatory activities, i.e., classical pathway DAA

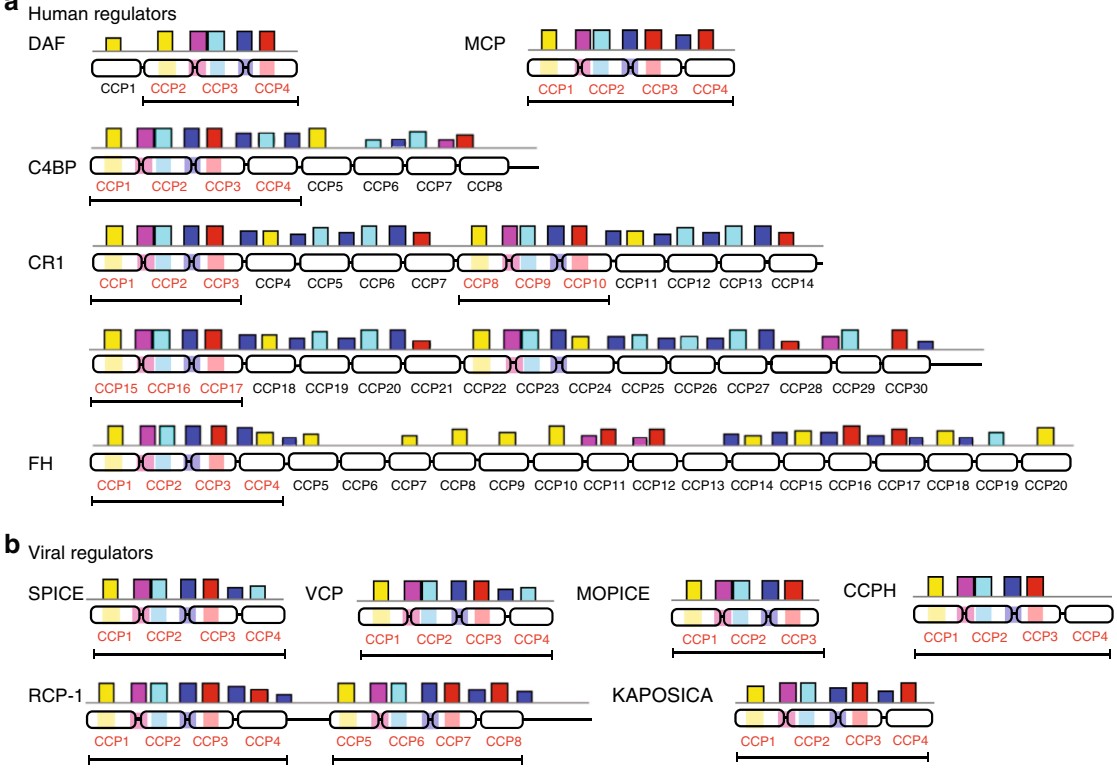

**Fig. 2** Human and viral complement regulatory domains contain a conserved signature motif pattern. MAST scanning showed the presence of a conserved motif pattern (vertical coloured bars; positional *p*-value < 0.0001) in human (**a**) and viral (**b**) complement regulatory domains (rounded rectangles). The CCP domains are numbered and regulatory CCPs are shown in red font. The location of signature motif pattern [M5 (yellow)-M3 (pink)-M1 (sky blue)-M2 (blue)-M4 (red)] is shown by shading on CCP domains with respective motif colours and the CCP domains implicated in function are marked by a horizontal line

(e.g., DAF CCP2-4, C4BP CCP1-3, CR1 CCP1-3), alternative pathway DAA (e.g., DAF CCP2-4, FH CCP1-3, CR1 CCP1-3), C3b CFA (e.g., MCP CCP1-3, CR1 CCP8-10 & 15–17 and FH CCP1-3) as well as C4b CFA (e.g., MCP CCP1-3, CR1 CCP8-10 & 15–17 and C4BP CCP1-3) (Fig. 2). Essentially similar results were also observed for the viral RCA regulators (Fig. 2).

Human RCA-like complement regulators are also conserved in non-mammalian vertebrates. Moreover, such proteins in chicken, zebrafish, European carp, Arctic lamprey and the barred sand bass have been shown to possess the complement regulatory function[29–33]. We thus looked for the presence of the conserved motif pattern in these proteins by MAST scanning using 5-motifs. The signature motif pattern was found in chicken and the barred sand bass, but not in the other species (Supplementary Fig. 4). We thus looked at a combination of motifs that can provide a better identification, i.e., combinations that can provide the maximum specificity with negligible false positives. A 2-motif pattern (M5–M3) provided the specificity of 0.64, a 3-motif pattern (M5-M3-M1) provided the specificity of 0.93, and a 4-motif pattern (M5-M3-M1-M2) provided the specificity of 0.99. Although the specificity of 0.93 is very close to 0.99 and 1.0 that was provided by 4-motif and 5-motif patterns, respectively, it included a few non-regulators, e.g., complement factor B, FH-related protein-2, C1s and presence of additional sites in the non-regulatory region of FH etc. (Supplementary Fig. 5). Hence, we concluded that for best results we must use a 4-motif scan. The 4-motif pattern (M5-M3-M1-M2) identified regulatory CCP domains in all the proteins, except chicken CREG (Supplementary Fig. 4). Interestingly, chicken, zebrafish and European carp proteins showed one regulatory site, whereas Arctic lamprey and barred sand bass proteins showed two regulatory sites. These

results, therefore, demonstrate a broad range sensitivity and specificity of the motif pattern in recognizing complement regulatory CCP domains in mammalian as well as non-mammalian vertebrate sequences.

To further ascertain the robustness of 4-motif pattern in determining the complement regulatory domains in mammalian RCA protein, we also scanned human and viral RCA sequences with these motifs. The 4-motif pattern remarkably identified the regulatory domains in all the proteins barring CR1. In CR1 it identified even the first three CCPs of LHR-D as the regulatory domains, which lack the regulatory function (Supplementary Figs. 6a and 6b). Thus, we inferred that the 5-motif pattern exhibits higher specificity in recognizing mammalian complement regulatory CCPs, while the 4 motif pattern exhibits higher specificity in identifying non-mammalian complement regulatory CCPs.

**Phylum-wide motif search reveals motif pattern until Cnidaria.** Encouraged by the identification of 4-motif pattern in non-mammalian complement regulatory CCPs with high specificity, we looked for the presence of such a motif pattern in CCP domain-containing proteins across all phyla. For this, we extracted the sushi domain-containing sequences from database (PF00084) which has a large collection of protein families that are Pfam annotated based on their domain architecture. These sequences were subjected to MAST scanning using 4-motifs. The motif pattern was found in all chordates including urochordates (Fig. 3 and Supplementary Data 3). Thus far, however, functional characterization of regulatory RCA proteins have been done only up to lamprey (Agnatha)[33]. Additionally, we also found the motif

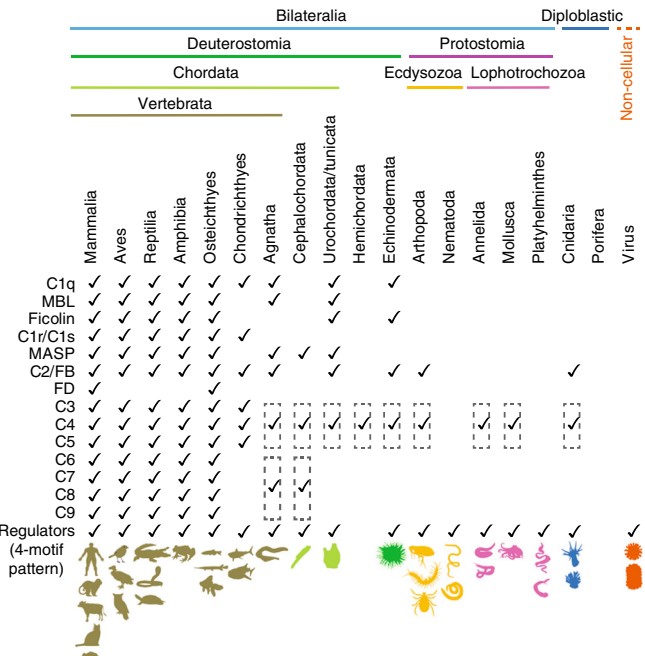

**Fig. 3** Phylum-wide search for the presence of signature motif pattern in RCA-like sequences. The presence of the sequence motif pattern was investigated in all the CCP containing sequences available in the Pfam database [pfam family: Sushi (PF00084)]. The presence/absence of motif-recognized regulators is compared with the presence/absence of complement components in each phylum[1]. The tick represents the presence of protein and the dashed box represents the presence of C3/C4/C5-like or C6/C7/C8/C9-like protein. A few model organisms/viruses in which regulator-like sequences are found are drawn below (a detailed list is provided in Supplementary Data 3)

pattern in animals belonging to phyla Echinodermata, Arthropoda, Nematoda, Annelida, Mollusca, Platyhelminthes and Cnidaria, suggesting that acquisition of complement regulatory activity is an ancient event (Fig. 3 and Supplementary Data 3). The presence of C3/C4/C5-like proteins has been reported till cnidarians[1], which explains the requirement of regulatory proteins in these phyla. It is interesting to note that nematodes and flatworms (Platyhelminths), which do not have C3/C4/C5-like proteins, also contain proteins with such a motif pattern.

**Loss of M4 in LHR-D of CR1 causes a loss in regulatory activity.** Among the human complement regulators, CR1 is the only regulator that encompasses three distinct regulatory sites—one in each of its LHRs. The fourth LHR or LHR-D, however, lacks the regulatory site and the reason for this is still unknown. As pointed out above, our examination of the LHRs for the presence of 5 motif pattern showed the absence of motif 4 (M4) in LHR-D (CCP22-24) as opposed to LHR-A, -B and -C (Figs. 2, 4a). Alignment of M4 of LHR-A, -B, and -C, and the homologous region of LHR-D showed that the LHR-D region differs in 11 amino acids compared to other LHRs (Fig. 4b). Notably, multiple residues (E633, H636, Y637, S639, V640 and R644) in M4 of LHR-B were shown to contribute to the C3b/C4b CFA and/or binding[34]. Further, the recent co-crystal structure of C3b with regulatory domains of CR1 (CCP15-17) showed M4 as a crucial site for C3b interaction[24]. These observations thus associated the lack of activity in LHR-D with the absence of M4 and raised the possibility of restoring the regulatory activity by motif substitution. Sequence analysis revealed that LHR-D domains CCP22-24

are more similar to LHR-A domains CCP1-3 than LHR-B domains CCP8-10 or LHR-C domains CCP15-17 (Supplementary Fig. 7). Consequently, we substituted the M4 of CCP3 (LHR-A) at the collinear site of CCP24 (LHR-D) and expressed this substitution mutant [LHR-DM4$^A$ (CCP22-24)] along with LHR-A (CCP1-3) and LHR-D (CCP22-24) domains (Fig. 4c and Supplementary Figs. 8 and 9).

In CR1, the regulatory site 1 (LHR-A (CCP1-3)) primarily display decay-accelerating activities for the classical and alternative pathway C3 convertases. We thus assessed LHR-DM4$^A$ (CCP22-24), LHR-A (CCP1-3) and LHR-D (CCP22-24) molecules for these activities. The LHR-DM4$^A$ (CCP22-24) molecule exhibited a gain in both these activities over LHR-D (CCP22-24). The gain, however, was more substantial for the classical pathway DAA compared to the alternative pathway DAA (Fig. 4d, e). We also compared the cofactor activities of LHR-DM4$^A$ (CCP22-24) for C3b and C4b with LHR-A (CCP1–3) and LHR-D (CCP22–24). The substitution mutant showed only a minimal gain in the cofactor activities (Supplementary Figs. 10a and 10b). To gain a better understanding of the interaction of M4 residues of LHR-A (CCP1–3) with the target protein C3b and explain why lack of M4 in LHR-D (CCP22–24) results in loss in the regulatory activities, we generated homology models of LHR-A (CCP1–3) and LHR-D (CCP22–24) with C3b, using the crystal structure of LHR-C (CCP15–17):C3b complex as a template (PDB ID 5fo9; Supplementary Fig. 10c). The model indicated loss of two important interactions (Fig. 4f) in CCP24 with C3b (N1540 with Q185 and Q1547 with H1312) as opposed to CCP3 (Y187 with Q185 and R194 with H1312). Thus, the motif-pattern partially unravelled the inexplicable reason for the loss of regulatory activity in LHR-D (CCP22–24) and illustrated the importance of each of the motifs in complement regulation.

**Signature motif pattern identifies polydom as complement regulator.** The success of identification of motif patterns led us to the exciting possibility of identifying novel complement regulators in humans. We, therefore, performed MAST scanning of human protein database (NCBI proteome ID:UP000005640) using 4 and 5 motifs. The search revealed the signature motifs in two novel proteins to our knowledge apart from the well-known complement regulators: (a) β$_2$-glycoprotein I, and (b) polydom/Svep1. Interestingly, both the proteins had the first 4 motifs (M5-M3-M1-M2) and the fifth motif (M4) was replaced by M1 (Supplementary Fig. 11). We would like to point out here that M4 is most similar (60% similarity) to M1 and the MAST search prefers a replacement of motifs that are ≥60% similar to each other. This, therefore, suggested that both these proteins are likely to have the complement regulatory activities. β$_2$-glycoprotein I (Apolipoprotein H) has recently been demonstrated as complement regulator[35]. Interestingly, the regulatory activity was shown to reside precisely where the motif pattern resides (i.e., CCP1-3). It, however, is an unusual complement regulator in that it binds C3 and recruits FH for its inactivation with the help of factor I[35]. Polydom (also known as SVEP1; Sushi, von Willebrand factor type A, EGF and pentraxin domain-containing protein 1) on the other hand is a member of the pentraxin family and is not yet associated with any specific function. It is a large protein (387 kDa) with a unique blend of domains including 34 CCP domains (Fig. 5a). Expression analysis of polydom showed that it is strongly expressed in human and mouse placenta[36] and in adult bone-associated skeletal tissues, mesenchymal stromal cells, and pre-osteoblastic cells[37].

To determine whether polydom indeed has complement regulatory activities, we tested its ability to function as a cofactor for factor I as well as decay the preformed C3 convertases. As the

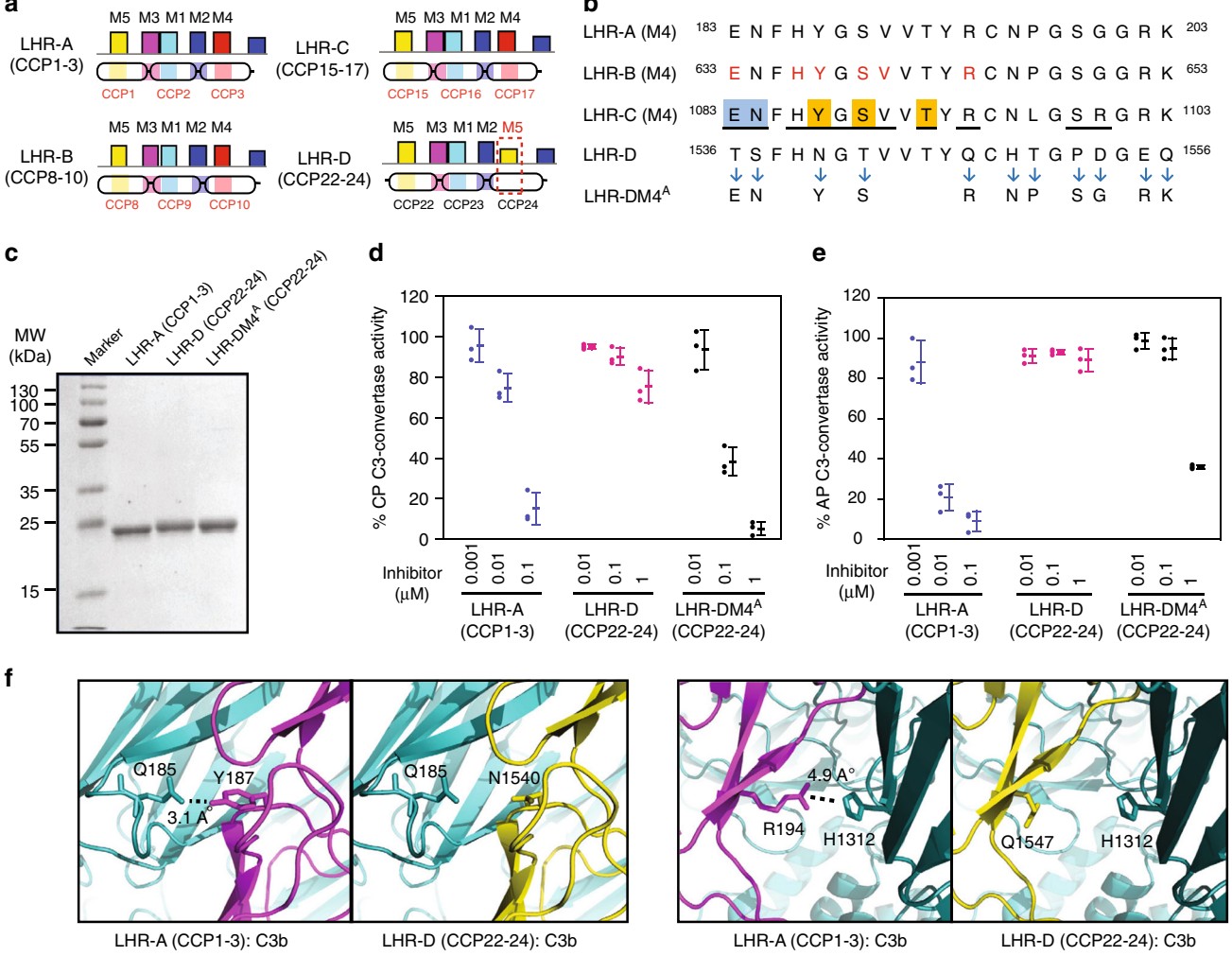

**Fig. 4** Inability of LHR-D (CCP22-24) in CR1 to function as a complement regulator stems from the loss of Motif 4 (M4). **a** MAST scanning of signature motifs in regulatory domains of long-homologous repeat (LHR)-A, -B, -C and -D. Loss of M4 in LHR-D (CCP22-24) is highlighted (red dashed box). **b** M4 motif sequences in LHR-A, -B, -C and -D are depicted alongside LHR-DM4$^A$ (LHR-D with M4 of LHR-A). The eleven amino acid residues that were changed in LHR-DM4$^A$ (CCP22-24) according to LHR-A (CCP1-3) are marked by arrows. Previously reported functionally important residues in LHR-B (CCP8-10) are represented in red font. The LHR-C (CCP15-17):C3b interface residues of M4 region are underlined (referred from reported crystal structure; PDB ID: 5fo9). The LHR-C (CCP15-17) residues involved in strong hydrophobic interactions, salt bridges or specific hydrogen-bonding with MG1 (blue) and MG2 (orange) domains of C3b are highlighted. **c** SDS-PAGE analysis of purified LHR-A (CCP1-3), LHR-D (CCP22-24) and LHR-DM4$^A$ (CCP22-24). **d, e** Classical pathway-decay-accelerating activity (**d**) and alternative pathway- decay-accelerating activity (**e**) of LHR-A (CCP1-3), LHR-D (CCP22-24) and LHR-DM4$^A$ (CCP22-24). C3-convertase activity in the absence of a regulator was considered as 100% and used for normalization. Data are presented as dot-plot with mean ± SD of three independent experiments. **f** Interaction mapping of LHR-A (CCP3) and LHR-D (CCP24) with C3b. Models of LHR-A (CCP1-3):C3b and LHR-D (CCP22-24):C3b complexes were constructed using the crystal structure of LHR-C (CCP15-17):C3b complex (PDB ID: 5fo9). LHR-D (CCP24) exhibited loss of two C3b interactions in the M4 region – at N1540 and Q1547 (collinear to Y187 and R194 of LHR-A, respectively)

motif pattern was found only in CCP12-14, we expressed this region of polydom, along with the accompanying C-terminal domain (CCP15), because in many human RCA proteins, the C-terminal domain assists in binding to C3b/C4b (Fig. 5b and Supplementary Fig. 8). The expressed protein displayed the CFA for C3b, but not C4b (Fig. 5c–e and Supplementary Fig. 9). It also did not show any DAA for the classical and alternative pathway C3 convertases (Supplementary Fig. 12). These findings thus reiterate that the presence of a signature motif pattern in complement regulators indicates the presence of binding sites for complement components and regulatory activities.

**Importance of the motifs in interaction with C3b, FI and Bb/C2a.** The RCA proteins impart their regulatory activities—CFA

and DAA—owing to the formation of trimolecular complexes[25,38,39]. During CFA, the RCA protein interacts with C3b/C4b and factor I, while during DAA the RCA protein interacts with C3b/C4b and Bb/C2a. It is, therefore, plausible that the motifs associated with the regulators are a part of the interfaces between RCA and its interacting partners. We thus mapped the motifs onto the available experimentally solved structure of complexes.

To understand the importance of motifs in the CFA, we mapped the motifs onto the recently solved crystal structure FH complexed with C3b and factor I (FI) (C3b-FH-FI complex; PDB ID: 5O35)[25]. The protein–protein interfaces were then estimated by calculating the buried surface area (BSA) using PISA (http://www.ebi.ac.uk/msd-srv/prot_int/pistart.html). Of the total interface between C3b and FH (BSA: 1586.3 Å²), the motifs covered

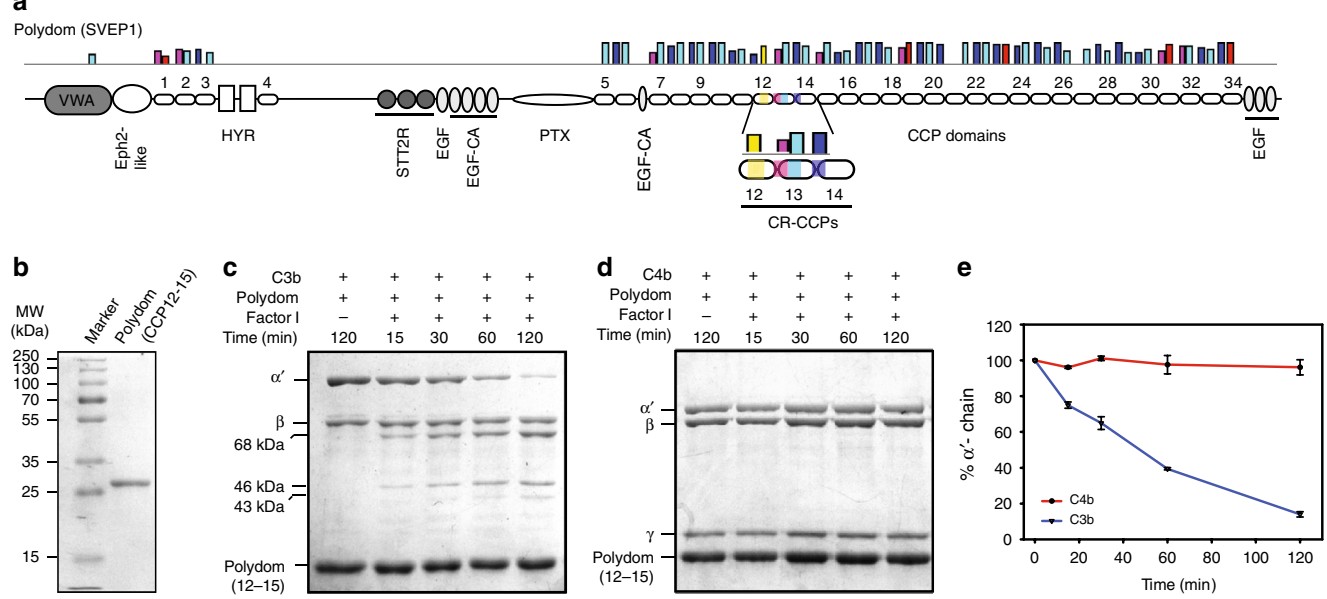

**Fig. 5** Identification of polydom as a complement regulator. **a** Schematic representation of polydom showing domain arrangements and the presence of motifs (M1–M5; vertical coloured bars) in CCP domains. The signature motif pattern was observed in CCP12-14 [marked with the horizontal bar as complement regulatory (CR)-CCPs]. **b** Resolution of purified polydom on SDS-PAGE **c**, **d** C3b-cofactor activity (**c**) and C4b-cofactor activity (**d**) of polydom. **e** Graphical representation of % α′-chain over time. Data are presented as mean ± SD of three independent experiments

62.4% of the area (BSA: 987.81 Å²), where M3 (that spanned the linker between CCP1-2) covered the most (BSA: 434.41 Å²; 27.4%) and displayed interaction with α′-NT, MG6, MG2 and MG7 domains of C3b (Fig. 6a). The other motifs that covered the interface include M4 (BSA: 212.7 Å²; 13.4%), M1 (BSA: 187.18 Å²; 11.8%) and M5 (BSA: 155.52 Å²; 9.8%). M4 showed interaction with MG2 domain, M1 exhibited interaction with MG6 and MG2 domains, and M5 displayed interaction with α′-NT and MG7 domains. Interestingly, M2 did not show interaction with any of the domains of C3b (Fig. 6a).

Next, we looked at the interface between FH and FI. The total interface between FH and FI was 1039.86 Å², of which, 75.5% (BSA: 784.95 Å²) was covered by the motifs. Here, M2, which spanned the linker between CCP2-3, majorly covered the interface (BSA: 448.2 Å²; 43.1%) and primarily interacted with the serine protease (SP) domain of FI (Fig. 6b). Besides M2, M1 (BSA: 202.66 Å²; 19.5%) and M4 (BSA: 134.06 Å²; 12.9%) also showed interaction with the SP domain of FI (Fig. 6b). M3 and M5 motifs did not show any interaction with FI.

To understand the contribution of motifs in DAA, we looked into the available mutagenesis data as the structure of RCA with C3 convertase (C3bBb or C4b2a) is not available. We observed that majorly mutations in M5 are linked with loss in DAA without any loss in binding to C3b/C4b suggesting that M5 is likely involved in the interaction with Bb/C2a (Supplementary Data 2). The availability of three-dimensional structures of RCA with C3 convertases will provide a better insight into the involvement of these motifs in DAA.

## Discussion

The RCA proteins are essential for preventing complement activation both on the cell surface and in the fluid-phase[40–42]. They are entirely composed of CCP domains; however, not all domains have regulatory activity rather a stretch of 3–4 CCPs harbour this function. Notably, here, we have identified five unique conserved motifs in RCA proteins, which provide ab initio prediction of regulatory domains in the RCA proteins.

To date, there is no method to predict regulatory RCA proteins. Therefore, predictions are typically made on the basis of the exclusive presence of CCP domains in a protein, and sequence similarity with the known RCA proteins. This, however, is not a sufficient criterion for such prediction as human complement receptor 2 (CR2) and vaccinia virus protein B5R, which are exclusively formed by CCP domains, lack complement regulatory function. Moreover, we also know that proteins which are not solely formed by CCP domains possess complement regulatory activity (e.g., human CSMD1[43]). Here, we have developed a multiple motif-based approach to identify regulatory RCA proteins. The string of motifs identified in the present study was clearly capable of identifying all the known regulatory RCA proteins, and more importantly, the regulatory domains within them, with only two exceptions (human CSMD1[43] and chicken CREG[32]). We suggest that the high sensitivity and specificity of our method is owing to extremely low E values of the identified motifs ($E = 2.6 \times 10^{-887}$ to $2.3 \times 10^{-706}$).

Of the five conserved motifs, three (M1, M4 and M5) are located around the second Cys of each of the three tandem regulatory CCP domains. Earlier, Barlow and colleagues have reported the presence of a large motif (hXhGXXhX hXC$^{II}$XXG↑hXhXG) around the second Cys of most CCP domains in RCA proteins[11]. They determined this by multiple sequence alignment of 84 individual CCP domains of six different RCA proteins. In contrast to the earlier study, in the present study, input sequences for motif search by MEME utilized only the regulatory and putative regulatory CCP domain sequences of a large ensemble of RCA proteins (85 proteins) and it resulted in the identification of unique motifs around second Cys of each of the regulatory CCP domains. The sequence similarity amongst motifs M1, M4 and M5 is ~50–60% suggesting subtle important variations in these motifs determine their uniqueness. It is obvious from the motif sequences that each of the motifs is studded with many conserved residues. A critical look at the previous mutagenesis data show that many conserved motif residues are indeed linked to the regulatory function, and this

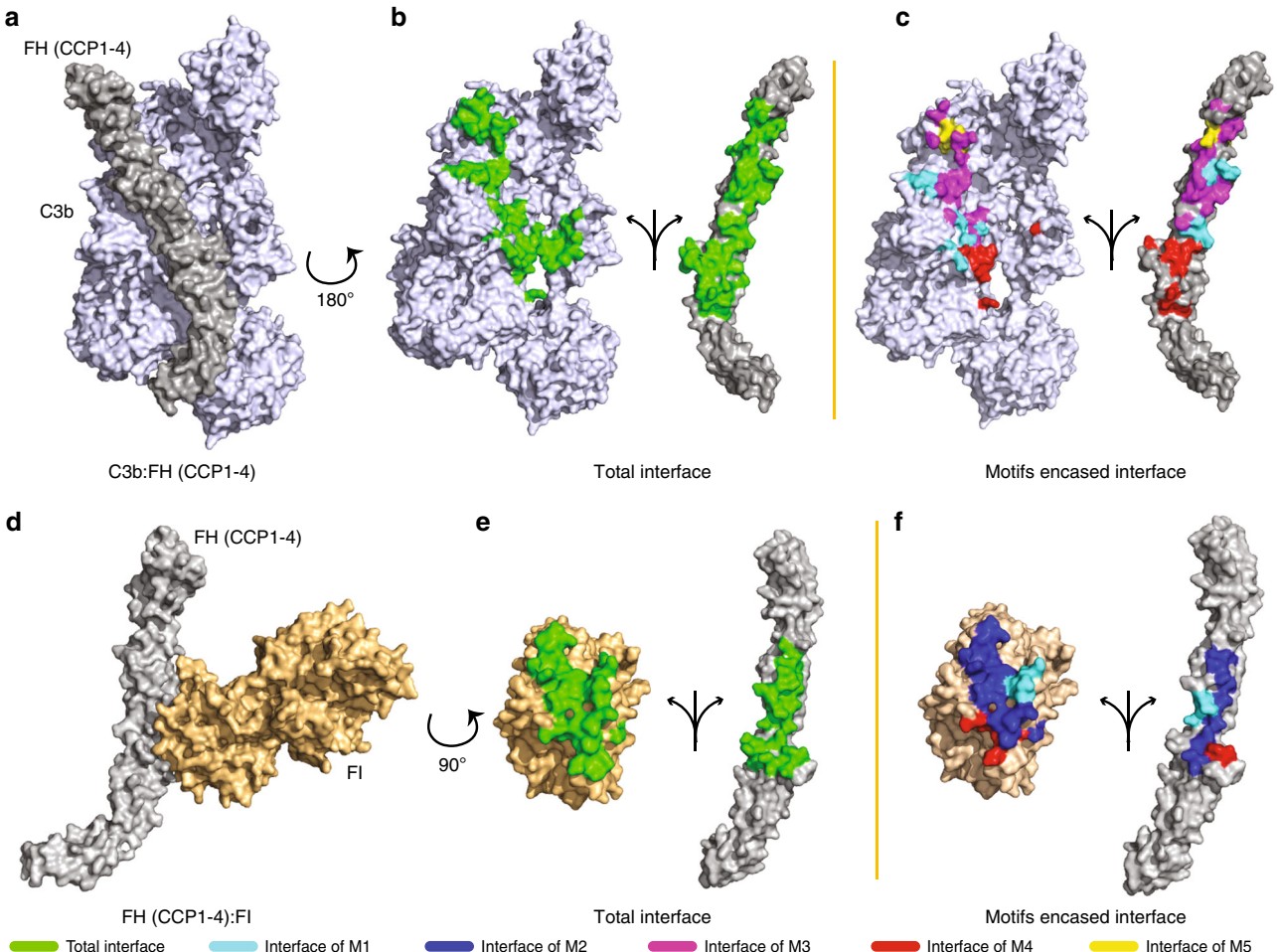

**Fig. 6** Mapping of motifs at FH-C3b and FH-FI interfaces. **a**, **d**, the complexes of FH with interacting partners (C3b and FI). **b**, **e**, the total interface is shown with green coloured residues. **c**, **f**, motif residues involved in the interface are shown with respective motif colours (identified below)

correlates well with the interface at which they are present (Supplementary Data 2). For example, motifs M3 and M4 largely cover the interface between RCA protein and C3b and mutations in the residues in these motifs are known to affect both CFA and DAA. The motif M2, on the other hand, is present at the interface between RCA and factor I and thus mutations in this motif primarily affect CFA. Our recent study shows that selective substitutions in the motif M2 of DAF-MCP chimera result in a substantial gain in CFA[44]. Interestingly, M5 might be involved in interaction between RCA and Bb/C2a as mutations in M5 residues affect only DAA.

Motif scanning in mammalian RCA proteins by MAST showed that all the identified motifs (M1–M5) are located in the regulatory unit of three CCP domains. However, in many known regulatory RCA proteins, the fourth successive domain plays a supportive role. We, therefore, also looked for the presence of the identified motifs in the fourth domain. Interestingly, in each case where the fourth domain participates in the function (e.g., C4BP, FH, MCP, SPICE, VCP, RCP-1 and Kaposica) there was a reappearance of two motifs—one positioned around the linker between CCP 3–4 and the other in CCP4. The motif at the CCP 3–4 linker was always M2, while the motif in the CCP4 was M4, M5 or M1. It is, therefore, likely that these motifs also participate in complement regulation.

Among poxviruses, RCA proteins (based on sequence similarity) are encoded by viruses belonging to genera Orthopoxvirus (e.g., vaccinia, variola, monkeypox, cowpox, ectromelia and

horsepox viruses), Suipoxvirus (e.g., swinepox virus), Leporipoxvirus (e.g., myxoma virus), Capripoxvirus (e.g., goatpox, sheeppox and Lumpy skin disease viruses), Cervidpoxvirus (e.g., deerpox) and Yatapoxvirus (e.g., yaba monkey tumour and Yaba-like disease virus)[9]. Importantly, the sequence similarity amongst them exceeds 91%. Earlier functional characterization of poxviral RCA proteins demonstrated that homologs of RCA proteins encoded by vaccinia (VCP[45]), smallpox (SPICE[46]), cowpox (IMP[47]) and monkeypox (MOPICE[48]) viruses have the capability to regulate complement. Whether other poxviral complement regulators are also functional is not known. The motif scan by MAST showed that horsepox and deerpox viruses have the required motif pattern, whereas the other pox viruses either have less number of motifs or the order is altered suggesting they are less likely to have the complement-regulatory function.

The complement system is ancient in origin. Initial studies based on the identification of complement components suggested that the system is restricted to vertebrates[49]. However, later, the presence of C3/C4/C5-like proteins was also found in invertebrates including sea urchin[50], horseshoe crab[51], and sea anemone[52]. MAST scanning for the presence of the 4-motif pattern in the sushi domain-containing sequences from the Pfam database showed that protein sequences from all the phyla, which contain C3/C4/C5-like protein, encompass this motif pattern. It thus implies that protostomes encode RCA-like proteins to regulate the primitive alternative and lectin pathways present in them[1]. It is imperative to point out here that the 4-motif pattern

was also found in nematodes (e.g., filarial worms and round-worms) and flatworms (e.g., tapeworm and flukes) which have not been reported to encode C3/C4/C5-like proteins. It would, therefore, be interesting to study whether these worms have acquired the RCA-like proteins from the host to subvert the host complement system as that seen in viruses; horizontal gene transfer has been reported in these organisms[53].

To experimentally demonstrate the efficiency of our motif pattern in identifying unknown regulatory RCA proteins, we show that the introduction of the lost motif (M4) in LHR-D (CCP22–24) recovers the DAA. The recovery, however, was near complete (~2.5-fold less) for the classical pathway DAA but limited (~135-fold less) for the alternative pathway DAA. It is, therefore, obvious that determinants of the later activity are also located elsewhere in the protein which is not a part of these motifs. One such example is Trp48 of LHR-A (CCP1–3), which is not a part of M1–M5, but is critically involved in the alternative pathway DAA; LDR-D (CCP22–24) contains Gln at this position[14]. Our results, therefore, reiterate the current belief that residues important for the regulatory function in the loop or insertion/deletion regions often do not come in long motifs.

Complement regulation is essential at the fetomaternal interface, and it is believed that such regulation is mediated by DAF, MCP and CD59, which are expressed at a high level on tropho-blast cells[54]. Previously, multiple complications such as preterm birth, fetal growth restriction and pregnancy loss have been linked to excessive complement activation[55,56]. Our MAST motif-scanning of human protein database showed the presence of 4-motifs (M5-M3-M1-M2) in polydom, which is also highly expressed in the placenta[36]. Expression and functional characterization of the CCP domains of polydom that encompass the regulatory motifs showed that it does possess the ability to inactivate C3b with the help of factor I, but the activity was moderate. We thus suggest that the presence of the 5-motif pattern in human proteins reflects the existence of optimum complement regulatory activity. Whether polydom contributes to complement regulation in the placenta, require further studies.

In summary, we have found five unique motifs, which when present in a specific order (M5-M3-M1-M2-M4) at specific locations in CCP proteins, have a strong predictive power for identification of regulatory CCP domains in human proteins. The predictive power of 4-motif pattern (M5-M3-M1-M2), however, is greater for identification of regulatory motifs in non-mammalian CCP containing proteins. Importantly, these motifs cover a large portion of the interfaces between RCA protein and its interacting partners such as C3b/C4b and FI, and likely Bb/C2a. Based on the presence of these motifs in CCP-containing proteins of animals across the phyla, we suggest that primitive alternative and lectin pathways present in protostomes are likely to be regulated by the RCA-like proteins. Owing to the importance of the motifs identified here, in predicting regulatory RCA proteins, we have developed an in silico regulatory RCA prediction tool CoReDo (Complement regulatory domains; http://coredo.nccs.res.in/meme-5.0.3/CoReDo/home.html) that allows scanning of the unannotated proteins for the presence of the regulatory motifs. We suggest that the use of this tool for the identification of putative regulatory RCA proteins/domains followed by their experimental validation would be an effective approach to identify unrecognized regulatory RCA proteins.

## Methods

**Selection of input sequences**. The sequences of RCA proteins of viruses and mammals for sequence motif search were selected based on a two-step approach. First, protein sequences were retrieved from NCBI (www.ncbi.nlm.nih.gov/) and UniProt (www.uniprot.org/) using BLAST where sequences of functionally characterized proteins (e.g., DAF, MCP, CR1 etc.) served as the query sequences. Next,

to discriminate between the regulatory and non-regulatory RCA sequences, the retrieved sequences were subjected to phylogenetic analysis by Neighbour-Joining (NJ) algorithm using MEGA5[57], and sequences that were falling only within a clade with functionally known sequence were considered (see Supplementary Fig. 1) and subjected to motif analysis. In the RCA proteins, typically consecutive 3–4 CCP domains form a functional unit and therefore phylogenetic analysis was performed only using sequences of successive 3–4 CCP domains. Because virus-encoded RCA proteins do not exceed four CCPs (except in RCP-1), their phylogenetic analysis was performed using the full protein sequences. However, in mammalian sequences, the number of CCP domains in a single chain varies from 4 to ~30. Hence, as an extra step was added to select the putative four regulatory CCPs for the phylogenetic analysis. Thus, phylogenetic tree of individual CCPs (i.e., CCP1, CCP2, and so on) was constructed, and sequences wherein all the individual CCPs that form a clade with respective functionally characterized human CCPs (e.g. CCP1 with CCP1 and so on) were selected for further analysis (see Supplementary Fig. 2). Additionally, for reducing the redundancy, sequences which showed more than 95% similarity were removed and for unbiased motif construction, sequences representing each type of RCA regulator (e.g., DAF-like, CR1-like, C4BP-like, MCP-like, FH-like, poxvirus-like and herpesvirus-like; Supplementary Data 1) were included with approximately equal weightage in the input sequences. In all, a total of 85 sequences (Supplementary Data 1) were finalized for identification of sequence motifs.

**Motif search by MEME and motif scanning by MAST**. Motif search in the selected sequences was conducted by MEME[58] (http://meme.nbcr.net). The width parameter of motifs that was selected was 18–21 as it was most apt in differentiating regulatory and non-regulatory sequences. As input parameters, "one occurrence per sequence" was selected for searching a minimum five motifs. Because a minimum of three sequential CCPs are required for forming a functional unit (e.g., in DAF, CR1 and viral RCAs), the input sequences for motif search were of only three consecutive CCPs. Motif scanning was performed by Motif Alignment and Search Tool (MAST) available in the MEME suite.

**CoReDo: A tool for predicting regulatory RCA proteins**. Complement Regulatory Domain (CoReDo) is a simple and efficient tool for prediction of complement regulatory RCA proteins. The web server accepts protein sequence in FASTA format as input. The submitted sequence is scanned against the set of motifs (either on the basis of four motifs or five motifs) mentioned before using Motif Alignment and Search Tool (MAST)[59]. Simultaneously, to identify the functionally characterized protein domains, the sequences are scanned against SMART[60] protein domain database through InterPro[61] web server. The arrangement(s) of the motifs is/are scanned by using in-house Perl scripts which can also indicate the regulatory site, if present, in the given sequence. A graphical representation, as well as the tabular output indicating positions of motifs/domains, are summarized in the output page. The web server was designed using PHP language using XAMPP server. It is publicly available at http://coredo.nccs.res.in/meme-5.0.3/CoReDo/home.html.

**Calculation of sensitivity and specificity**. A total of 162 sequences were taken for the calculation of sensitivity and specificity. These sequences encompassed 122 functionally characterized (or phylogenetically related as explained in input sequences for motif generation) regulatory CCPs or functional units (positive hits) and 129 functionally characterized non-regulatory CCPs or non-functional units (negative hits). For example, FH has 20 CCP domains, with one positive hit that ought to be recognized by the motif-pattern (i.e., CCP1-3), and 17 negative hits (i.e., CCP2–4, 3–5, 4–6, 5–7, 6–8, 7–9, 8–10, 9–11, 10–12, 11–13, 12–14, 13–15, 14–16, 15–17, 16–18, 17–19 and 18–20) that must not be recognized by the motif-pattern. The true positives (TP), true negatives (TN), false positive (FP) and false negative (FN) were defined as below. The true positives (TP) were the positive hits that were detected correctly by the motif pattern, and true negatives (TN) were the negative hits that were not detected by the motif pattern. Further, the false negatives (FN) were the positive hits that the motif pattern failed to detect and the false positives (FP) were the negative hits that were falsely identified as positive by the motif pattern. The sensitivity and specificity were then calculated according to the following equations.

$$Sensitivity = TP/(TP + FN)$$

$$Specificity = TN/(TN + FP)$$

Thus, the sensitivity stated how often a motif(s) was successfully detected in regulatory sequences, and specificity stated how efficiently a motif was detected only in regulatory sequences and not in the non-regulatory sequences.

**Phylum-wide search for the presence of sequence motif pattern**. The presence of sequence motif pattern that annotates complement regulatory CCP domains of RCA proteins was investigated in all the CCP containing sequences available in Pfam database. A total of 23,751 sequences that contain CCP domains were downloaded Pfam [Pfam family: Sushi (PF00084)[62]]. Next, the redundant sequences were removed using CD-Hit[63] and 5078 sequences were analyzed by

4-motif and 5-motif pattern by MAST. Information about the presence of other complement proteins was obtained from Nonaka et al.[1].

**Cloning and expression of LHR-A, LHR-D and LHR-DM4[A].** To generate CR1 cDNA, RNA was isolated by Trizol method from THP-1 cells (human monocytic leukemic cell line; National Centre for Cell Science, Pune) and converted to cDNA using High-Capacity cDNA Reverse Transcription Kit (Applied Biosystems, USA). The CR1 constructs, LHR-A (CCP1–3) and LHR-D (CCP22–24), were then amplified from cDNA using high fidelity DNA polymerase (Roche, country of origin) using the specific primers (Integrated DNA Technologies, Inc., Singapore) listed in Supplementary Table 1. For generation of LHR-DM4[A] (CCP22-24 with motif 4), LHR-D (CCP22–24) was amplified with two sets of primers (Supplementary Table 1): the first set amplified the region from the start of CCP22 till the region of motif 4 with overhanging motif region 4 of LHR-A (CCP1–3) in the reverse primer, and the second set amplified the region after motif 4 of CCP24 with overhanging motif region of LHR-A (CCP1–3) in the forward primer. The amplified products were annealed and amplified to obtain LHR-DM4[A]. The PCR amplified LHR-A (CCP1–3), LHR-D (CCP22–24) and LHR-DM4[A] (CCP22–24 with motif 4) were then cloned in pGEMT and sub-cloned in pET28. All the clones were validated by sequencing (1st Base Laboratories Sdn Bhd, Malaysia).

For expression, all the CR1 constructs cloned in pET28 were transformed into *E. coli* BL21 (DE3) cells. These cells were then grown in Luria-Bertani medium with kanamycin (25 μg/ml, final concentration) and protein expression was induced using 1 mM isopropyl-thio-D-galactopyranoside (IPTG; Sigma-Aldrich) as described[19,64]. The expressed protein was present in the inclusion bodies and hence was purified over nickel nitrilotriacetic acid-agarose (Ni-NTA) column (Qiagen) in the presence of urea. The eluted protein was refolded using the rapid dilution method standardized earlier in our laboratory[65] and loaded onto Superose 12 column (GE Healthcare Life Sciences) in phosphate-buffered saline (10 mM sodium phosphate and 145 mM sodium chloride, pH 7.4) to obtain a monodispersed population of the expressed protein. The purity of proteins exceeded 95% as judged by its analysis on 12% SDS-PAGE. The quality of protein and protein folding was checked by running them on SDS-PAGE under reducing and non-reducing conditions and subjecting them to circular dichroism (CD) analysis on Jasco J18 spectropolarimeter. All the expressed proteins showed slightly faster mobility on SDS-PAGE under non-reducing in comparison to the reducing conditions—an indication of disulfide bond formation. They also showed a peak around 230 nm, which is a characteristic feature of CCP domains[66].

**Alternative pathway DAA assay (AP-DAA).** The alternative pathway decay-accelerating activity of LHR-A (CCP1–3), LHR-D (CCP22–24) and LHR-DM4[A] (CCP22–24) was assessed by measuring the decay of $Ni^{++}$ stabilized AP C3-convertase (C3bBb) formed on rabbit erythrocytes using purified C3, factor B, and factor D in GVB buffer (gelatin veronal buffer; 5 mM barbital, 145 mM NaCl and 0.1% gelatin, pH 7.4)[28]. Herein, erythrocytes coated with C3bBb were incubated with or without increasing concentrations of each of the CR1 constructs at 37 °C for 10 min. Following this, the remaining C3-convertase activity was assayed by incubating the cells at 37 °C for 20 min with normal human sera containing 20 mM EDTA (NHS-EDTA; source of C3–C9) and measuring lysis. Data obtained were normalized by considering the lysis in the absence of inhibitor [LHR-A (CCP1–3) and LHR-D (CCP22–24) or LHR-DM4[A] (CCP22–24)] as 100% lysis. The $IC_{50}$ (50% of inhibitory concentration) was calculated graphically by plotting the normalized percent of lysis against inhibitor concentration. Use of human serum for the study was approved by the Institutional Ethical Committee of the National Centre for Cell Science, Pune (NCCS); informed consent was obtained from the subjects. Use of rabbit RBCs was approved by the Institutional Animal Ethics Committees of NCCS; they were obtained from the in-house animal facility of NCCS.

**Classical pathway DAA assay (CP-DAA).** The classical pathway decay-accelerating activity of LHR-A (CCP1–3), LHR-D (CCP22–24) and LHR-DM4[A] (CCP22-2) was measured by examining the decay of the CP C3-convertase (C4b2a) formed on antibody (ICN Biomedical Inc., Irvine, CA; Cat#55806; lot#03176; 1:80 dilution) coated sheep erythrocytes (EAs) by sequential addition of purified complement proteins C1, C4 and C2 in DGVB[++] buffer (dextrose GVB; 2.5 mM barbital, 73 mM NaCl, 0.1% gelatin, and 2.5% dextrose, pH 7.4 containing 0.5 mM $MgCl_2$ and 0.15 mM $CaCl_2$)[67]. In brief, sheep erythrocytes coated with C4b2a were incubated with or without increasing concentrations of each of the CR1 constructs and the enzyme was allowed to decay at 22 °C for 5 min. Thereafter, the remaining C3 convertase activity was assayed by incubating the cells at 37 °C for 20 min with guinea pig serum containing 20 mM EDTA (GPS-EDTA; source of C3–C9) and measuring lysis. Data were normalized by considering the lysis in the absence of inhibitor as 100% lysis. The $IC_{50}$ (50% of inhibitory concentration) was calculated graphically by plotting the normalized percent of lysis against inhibitor concentration. Use of guinea pig serum and sheep RBCs was approved by the Institutional Animal Ethics Committees of NCCS. The guinea pig serum was obtained from the in-house animal facility of NCCS and sheep RBCs were obtained from the local slaughterhouse.

**Cloning, expression and purification of polydom CCP12–15.** cDNA synthesized from RNA isolated from human bone marrow stromal cells (a kind gift from Dr. Mohan Wani, National Centre for Cell Science) was used as a template for the generation of polydom CCP12–15 construct. For RNA isolation, the stromal cells were spun down and resuspended in Trizol (Gibco). Following the addition of 200 μl chloroform, the cells were vortexed for 30 s and kept at room temperature for 10–15 min. Thereafter, the cells were centrifuged at 12,000 rpm for 15 min at 4 °C, and the aqueous layer was transferred to a new microcentrifuge tube. It was then gently mixed with 500 μl isopropanol and kept at room temperature for 15 min. The mix was again centrifuged at 12,000 rpm for 15 min, and the pellet was washed with 70% DEPC ethanol. The RNA pellet was then air dried, dissolved in 20 μl DEPC water, and utilized for cDNA synthesis using High-Capacity cDNA Reverse Transcription Kit (Applied Biosystems, USA). Next, the polydom CCP12–15 was amplified by PCR using the specific primers (Supplementary Table 1) and cloned into pGEMT. Following digestion with NdeI and HindIII, it was recloned into pET29 vector for expression. The positive clone was sequenced for validation (1st Base Laboratories Sdn Bhd, Malaysia).

For expression, polydom cloned in pET29 was transformed into *E. coli* BL21 (DE3) cells. The expression, refolding and purification of polydom was same as that described for CR1 constructs.

**Cofactor activity assay (CFA).** The CFA of expressed polydom was examined for C3b as well as C4b[19]. Briefly, 12 μg of C3b or C4b was mixed with 9 μg of polydom and 600 ng of factor I in a total volume of 90 μl in phosphate buffer saline (pH 7.4). This reaction mix was then incubated at 37 °C and aliquots of 15 μl were removed at various time points. The reaction was stopped by adding dithiothreitol. All the samples were then run on 10% SDS-PAGE gel for separation of C3b/C4b cleavage fragments, which were visualized by Coomassie blue staining. The percentage of C3b/C4b cleaved was quantitated by densitometric analysis (QuantityOne, Bio-Rad) of the α'-chain which was normalized to the β-chain (loading control). The C3b and C4b CFA of LHR-A (CCP1–3), LHR-D (CCP22–24) and LHR-DM4A (CCP22–24) were essentially performed as that for polydom except that the reaction mixtures were incubated for a fixed period (1 h) with variable concentrations of the regulators. The concentrations of LHR-A (CCP1–3) varied from 0.1 to 1.6 μM, and that of LHR-D (CCP22–24) and LHR-DM4A (CCP22-24) varied from 2.5 to 40 μM.

**Homology modelling of LHR-A (CCP1–3) and LHR-D (CCP22–24).** The models of LHR-A (CCP1–3):C3b and LHR-D (22–24):C3b complexes were done by using the available crystal structure of LHR-C (CCP15-17):C3b complex as template (PDB ID: 5FO9). Sequence identity values for LHR-A (CCP1–3) and LHR-D (22–24) against LHR-C (CCP15–17) are 75% and 59%, respectively. The homology modelling was performed on Discovery Studio v 3.5[68] (Dassault Systèmes BIOVIA 2016) using modeller ver 9[69]. Among the five generated models, each model was further refined by energy minimization using steepest descent method and best model was selected on the basis of DOPE score. The stereochemical quality of predicted model was evaluated using PROCHECK[70].

**Statistics and reproducibility.** Data are presented as mean ± SD, and experiments have been repeated three times.

**Reporting summary.** Further information on research design is available in the Nature Research Reporting Summary linked to this article.

## Data availability
All data supporting the findings of the present study are included in the published article and its supplementary information file. Raw data for Fig. 4c and Fig. 5e is included as supplementary data (Supplementary Data 4).

## Code availabiltiy
CoReDo is publicly available at http://coredo.nccs.res.in/meme-5.0.3/CoReDo/home.html.

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

## Acknowledgements

We thank Prof. John P. Atkinson (Washington University School of Medicine, USA) and Dr. Jayati Mullick (National Institute of Virology, Pune) for valuable suggestions and critical reading of the manuscript and Dr. Mohan Wani (National Centre for Cell Science, Pune) for providing human bone marrow stromal cells for the generation of cDNA. The authors also thank Gaurang Mahajan (National Centre for Cell Science, Pune), Mrs. Smita Saxena (Bioinformatics Centre, S. P. Pune University) and Dr. Anirban Dutta (Tata Research Development and Design Centre, Pune) for their help/suggestions in writing the script and Rajesh Solanki (National Centre for Cell Science, Pune) for his assistance in hosting the CoReDo tool. This work is done in partial fulfilment of the Ph. D. thesis of H.O. to be submitted to the S.P. Pune University. The authors acknowledge financial assistance from the Department of Biotechnology, New Delhi in the form of fellowships to H.O. and H.S.P. This work was supported by the Department of Biotechnology, India Project Grant BT/PR28506/MED/29/1307/2018 (to A.S.).

## Author contributions

A.S., S.C.M. and H. O. designed research; H.O., P.G., H.S.P., R.S. and A.G. performed research; H.O. and P.G. analyzed data; and H.O., P.G. S.C.M. and A.S. wrote the paper.

## Additional information

**Competing interests:** The authors declare no competing interests.

