## [Peer Review File · Communications Biology]

Reviewers' comments:

Reviewer #1 (Remarks to the Author):

The manuscript from Ojha et al. presents an interesting investigation of RCA proteins based on analysis of amino acid signatures and interpretation of the significance of these signature patterns towards complement regulation. The authors claim to have identified a sequence of short sequence patterns that distinguishes RCA molecules from CCP-containing proteins without RCA function. The proposed results are supported by experimental evidence on CR1: the authors managed to introduce regulatory activity to CCP22-24, a known non-regulatory pattern of this macromolecule. The authors implemented their bioinformatic tool on a web server (CoReDo), offering immediate application to the scientific community.

Overall, this is a well-written and interesting manuscript, and I find it appropriate for this journal provided that the authors address the following issues:

1) As stated by the authors, the signature proposed for mammalian RCAs encompasses five motifs in a specific order (M5-M3-M1-M2-M4). This signature is present in all known RCAs and is not present in other proteins bearing CCP domains. In this respect, I find the results presented at page 12 of the manuscript (section "Identification of novel human complement regulator proteins using signature motif pattern") quite confusing: for both beta2-glycoprotein I and Polydom, the patterns identified with the 5-motifs scanning are M5-M3-M1-M2-M3 and not M5-M3-M1-M2-M4. At page 10, the authors report clear evidence for lack of the M4 signature as the cause of absent regulator activity for CR1 CCP22-24. Based on such statements, I would have expected beta2-glycoprotein I and Polydom to be classified as non-RCA. Surprisingly, in Fig. 5, the authors show experimental data for cofactor activity in Polydom CCP12-15. This appears in contradiction with the proposed five-motifs pattern for mammalian RCAs. No experimental studies are shown for beta2-glycoprotein I. It should be noted that although beta2-glycoprotein was previously suggested to be a complement regulator (Gropp et al, 2011), its activity and ability to bind to C3b could not be confirmed by others (Forneris et al, 2016). Based on the proposed analysis, I think that the authors should provide a stronger argument to justify the lack of the M4 pattern for these "new" complement regulators, otherwise I suggest to revise (or possibly remove) the content of this section of their manuscript and to include some cautionary statements next to the identification of regulatory CCP domains in their CoReDo tool when the M4 pattern is not present in the 5-motif search (as for the case of CR1 CCP22-24).

2) Did the authors use Polydom CCP12-15 or CCP12-14 or both? These two different constructs are described in both in text and in figures, however the materials and methods list only CCP12-15.

Reviewer #2 (Remarks to the Author):

Ojha et al. raises our understanding of the regulatory sites of the RCA proteins by identifying a signature pattern of 5 amino acid sequence motifs that are common to the CCP domains that mediate C3b/C4b binding, cofactor activity, and decay acceleration. The 5 motif signature, which was derived from mammalian and viral proteins, was used to identify all the major mammalian and viral RCA regulatory sites but was it was not sufficiently sensitive to identify more than a few non-mammalian RCA proteins.. Many more non-mammalian RCA proteins (plus one novel human protein) were identified when the analysis employed the more sensitive (but less selective) 4 motif pattern. The authors provide compelling validation to these signatures by comparing them to the known human RCA 3D complexes.

In my estimation these motif patterns could be used to further our understanding of complement regulation during viral pathogenesis and in other species and may be of value in assessing the potential impact of rare RCA variants on health and disease.

Specific comments/concerns:

1. The authors need to better explain how they chose the 4 motif signature for their non-mammalian scans. Did they try other combinations of motifs? The present explanation, "We thus used 4-motif pattern by virtue of MAST preferred replacement of M4 by M1" was not clear to this reader.

2. Brodbeck et al. [J Immunology (1996) 56:2528-33] showed that DAF constructs that lack CCP 4 (and thus lack motif 4 and part of motif 2) have strong CP decay acceleration activity. This appears consistent with the finding that motif 2 binds to factor I in the FH/C3b/FI complex.

3. The authors modify the functionally inactive CR1 CCP 22-24 by introducing motif 4 in CCP 24 using 11 amino acid substitutions, thus completing the 5 motif RCA signature. The resulting recombinant construct has decay acceleration activity, especially for the CP convertase. Does the construct have C3b and/or C4b cofactor activity?

Point-by-point response to the reviewers' comments

Reviewer #1

General comments: The manuscript from Ojha et al. presents an interesting investigation of RCA proteins based on analysis of amino acid signatures and interpretation of the significance of these signature patterns towards complement regulation. The authors claim to have identified a sequence of short sequence patterns that distinguishes RCA molecules from CCP-containing proteins without RCA function. The proposed results are supported by experimental evidence on CR1: the authors managed to introduce regulatory activity to CCP22-24, a known non-regulatory pattern of this macromolecule. The authors implemented their bioinformatic tool on a web server (CoReDo), offering immediate application to the scientific community.

Overall, this is a well-written and interesting manuscript, and I find it appropriate for this journal provided that the authors address the following issues.

Response: We express our sincere gratitude to the reviewer for appreciating our work.

Specific comments/concerns:

1. As stated by the authors, the signature proposed for mammalian RCAs encompasses five motifs in a specific order (M5-M3-M1-M2-M4). This signature is present in all known RCAs and is not present in other proteins bearing CCP domains. In this respect, I find the results presented at page 12 of the manuscript (section "Identification of novel human complement regulator proteins using signature motif pattern") quite confusing: for both beta2-glycoprotein I and Polydom, the patterns identified with the 5-motifs scanning are M5-M3-M1-M2-M3 and not M5-M3-M1-M2-M4. At page 10, the authors report clear evidence for lack of the M4 signature as the cause of absent regulator activity for CR1 CCP22-24. Based on such statements, I would have expected beta2-glycoprotein I and Polydom to be classified as non-RCA. Surprisingly, in Fig. 5, the authors show experimental data for cofactor activity in Polydom CCP12-15. This appears in contradiction with the proposed five-motifs pattern for mammalian RCAs. No experimental studies are shown for beta2-glycoprotein I. It should be noted that although beta2-glycoprotein was previously suggested to be a complement regulator (Gropp et al, 2011), its activity and ability to bind to C3b could not be confirmed by others (Forneris et al, 2016). Based on the proposed analysis, I think that the authors should provide a stronger argument to justify the lack of the M4 pattern for these "new" complement regulators, otherwise I suggest to revise (or possibly remove) the content of this section of their manuscript and to include some cautionary statements next to the identification of regulatory CCP domains in their CoReDo tool when the M4 pattern is not present in the 5-motif search (as for the case of CR1 CCP22-24).

Response: The reviewer correctly pointed out that based on the data provided for CR1 LHR-D (CCP22-24) in Fig. 4, lack of M4 would mean the absence of any regulatory activity, but we show the presence of cofactor activity in Polydom CCP12-15 that lacks M4. We thank the reviewer for bringing up this important point.

The reason as to why we thought that Polydom might have the regulatory activity is as follows. In Polydom, motifs M5-M3-M1-M2 are followed by motif M1 which is 60% similar to motif M4. A similar replacement of M4 by M1 is also seen in beta2-glycoprotein I, which was reported to have regulatory activity (Gropp et al., 2011). Moreover, the MAST search prefers a replacement of motifs that are $\geq 60\%$ similar to each other (PMID: 9672829). In the case of CR1 LHR-D (CCP22-24) however, motifs M5-M3-M1-M2 are followed by M5 and not M1, and the similarity between these two motifs

is only 51%. We have now added the reasoning for the possible presence of regulatory activity in Polydom (Page 8, lines 14-19). As suggested, we have also added a cautionary statement in CoReDO tool regarding the presence of complement regulatory activity in proteins where M4 is replaced by M1.

We agree with the reviewer that although beta2-glycoprotein I was suggested to be a complement regulator (Gropp et al., 2011), its ability to bind to C3b could not be confirmed by Forneris et al., (2016). We, however, believe that the lack of binding of β 2-GPI to C3b could be due to the presence of a glycan at Asn162, a site which is important for interaction with C3b. This possibility was also suggested by Forneris et al.

2. Did the authors use Polydom CCP12-15 or CCP12-14 or both? These two different constructs are described in both in text and in figures, however the materials and methods list only CCP12-15.

Response: We sincerely apologise for the error in Fig. 5, Supplementary Fig. 8 and the text. We have used only CCP12-15 construct, and the same has now been depicted in Fig. 5 and Supplementary Fig. 8, and described in the text (Page 8, lines 33-36). We included CCP15 in the Polydom because, in many human complement regulators, the 4th domain is critical for optimal binding to C3b/C4b.

Reviewer #2

General comments: Ojha et al. raises our understanding of the regulatory sites of the RCA proteins by identifying a signature pattern of 5 amino acid sequence motifs that are common to the CCP domains that mediate C3b/C4b binding, cofactor activity, and decay acceleration. The 5 motif signature, which was derived from mammalian and viral proteins, was used to identify all the major mammalian and viral RCA regulatory sites but it was not sufficiently sensitive to identify more than a few non-mammalian RCA proteins. Many more non-mammalian RCA proteins (plus one novel human protein) were identified when the analysis employed the more sensitive (but less selective) 4 motif pattern. The authors provide compelling validation to these signatures by comparing them to the known human RCA 3D complexes.

In my estimation these motif patterns could be used to further our understanding of complement regulation during viral pathogenesis and in other species and may be of value in assessing the potential impact of rare RCA variants on health and disease.

Response: We express our deepest thanks to the reviewer for appreciating our study.

Specific comments/concerns:

1. The authors need to better explain how they chose the 4 motif signature for their non-mammalian scans. Did they try other combinations of motifs? The present explanation, "We thus used 4-motif pattern by virtue of MAST preferred replacement of M4 by M1" was not clear to this reader.

Response: We sincerely apologise for the lack of clarification on using 4-motif signature pattern for non-mammalian RCA scans. We did try 2 as well as 3 motif combinations, but this resulted in lower specificity (increase in false positives; data now included in Fig. 1d). The combination of 2-motif (M5-M3) provided the specificity of 0.64 whereas the combination of 3-motif (M5-M3-M1) provided the specificity of 0.93. Although the specificity of 0.93 is quite close to 0.99 and 1.0 that was provided by the 4-motif and 5-motif patterns, respectively, it included a few non-regulators, e.g., complement factor B, FH-related protein-2, two sites in the non-regulatory region of FH etc. (see new

Supplementary Fig. 5). Hence, we concluded that for best results we must use a 4-motif scan. We have now clarified this in the text (Page 6, lines 27-36).

2. Brodbeck et al. [J Immunology (1996) 56:2528-33] showed that DAF constructs that lack CCP 4 (and thus lack motif 4 and part of motif 2) have strong CP decay acceleration activity. This appears consistent with the finding that motif 2 binds to factor I in the FH/C3b/FI complex.

Response: We agree with the reviewer. We, however, would like to point out that the mere presence of motif M2 is not an absolute indicator of factor I interaction site as this motif is also present in DAF which lacks cofactor activity. In our recent study, we have shown that a few residues that actually reside in the M2 region dictate factor I interaction (Panwar et al., 2019, PNAS; <https://doi.org/10.1073/pnas.1818573116>).

3. The authors modify the functionally inactive CR1 CCP 22-24 by introducing motif 4 in CCP 24 using 11 amino acid substitutions, thus completing the 5 motif RCA signature. The resulting recombinant construct has decay acceleration activity, especially for the CP convertase. Does the construct have C3b and/or C4b cofactor activity?

Response: We had only looked at the gain in DAA in LHR-D mutant that was substituted with motif M4 [LHR-DM4^A (CCP22-24)] because this motif was from LHR-A, which is known to have good decay acceleration activity and reduced cofactor activity. However, we have now examined the C3b and C4b cofactor activity of the substituted mutant [LHR-DM4^A (CCP22-24)]. It showed only a minimal gain in cofactor activity compared to LHR-A (CCP1-3). These data have been included in Supplementary Fig. 9 and text (Page 7, lines 47-50).

Note: We would like to inform the reviewers that while generating a dataset of regulator-like sequences, we sought to remove sequences that showed >95% similarity, but 6 such sequences were inadvertently left in the dataset. Although this did not result in any change in the motif pattern in the regulatory domains of RCA proteins, there were minor changes in the motif pattern in the non-regulatory CCP domains of a few proteins. We apologize for this mistake. We have now incorporated these minor changes in the figures.

REVIEWERS' COMMENTS:

Reviewer #1 (Remarks to the Author):

The authors have carefully addressed my comments in the revised version of their manuscript. The reasoning added to the text is satisfactory and makes this version of the manuscript suitable for publication in my view. The new computational tool will definitely be useful to many in complement research.

Reviewer #2 (Remarks to the Author):

The authors have addressed the reviewers' concerns appropriately.